# Mutation in the Common Docking Domain Affects MAP Kinase ERK2 Catalysis and Stability

**DOI:** 10.3390/cancers15112938

**Published:** 2023-05-26

**Authors:** Leonore Novak, Maria Petrosino, Alessandra Pasquo, Apirat Chaikuad, Roberta Chiaraluce, Stefan Knapp, Valerio Consalvi

**Affiliations:** 1Dipartimento di Scienze Biochimiche “A. Rossi Fanelli”, Sapienza University of Rome, 00185 Rome, Italy; leonore.novak@hotmail.com (L.N.); valerio.consalvi@uniroma1.it (V.C.); 2Chair of Pharmacology, Section of Medicine, University of Fribourg, CH-1700 Fribourg, Switzerland; maria.petrosino@unifr.ch; 3ENEA CR Frascati, Diagnostics and Metrology Laboratory FSN-TECFIS-DIM, 00044 Frascati, Italy; alessandra.pasquo@enea.it; 4Institute of Pharmaceutical Chemistry, Goethe-University Frankfurt, 60438 Frankfurt, Germany; chaikuad@pharmchem.uni-frankfurt.de (A.C.); knapp@pharmchem.uni-frankfurt.de (S.K.); 5Structural Genomics Consortium, BMLS, Goethe-University Frankfurt, 60438 Frankfurt, Germany

**Keywords:** single nucleotide variant, ERK2, MAPK1, protein stability, mutation, MAP kinase, cancer mutations and protein structure alterations

## Abstract

**Simple Summary:**

The extracellular-signal-regulated kinase 2 (ERK2) plays a key role in the Ras-Raf-MEK-ERK signal transduction cascade and is involved in the regulation of many cellular processes. The ERK2 signaling converts extracellular stimuli into cell proliferation and, when deregulated, can promote oncogenic transformation. The ERK2 missense variants, carrying a single amino acid substitution in the protein sequence, have been identified in cancer tissues. This study reports a comprehensive biochemical and biophysical study of the ERK2 wild-type and variants in the common docking site present in cancer. A detailed analysis, from a molecular point of view, of the variants clarifies the impact of a single nucleotide variation on protein structure, stability and function, is essential to design alternative therapeutic strategies, and is a preliminary step to personalized medicine.

**Abstract:**

The extracellular-signal-regulated kinase 2 (ERK2), a mitogen-activated protein kinase (MAPK) located downstream of the Ras-Raf-MEK-ERK signal transduction cascade, is involved in the regulation of a large variety of cellular processes. The ERK2, activated by phosphorylation, is the principal effector of a central signaling cascade that converts extracellular stimuli into cells. Deregulation of the ERK2 signaling pathway is related to many human diseases, including cancer. This study reports a comprehensive biophysical analysis of structural, function, and stability data of pure, recombinant human non-phosphorylated (NP-) and phosphorylated (P-) ERK2 wild-type and missense variants in the common docking site (CD-site) found in cancer tissues. Because the CD-site is involved in interaction with protein substrates and regulators, a biophysical characterization of missense variants adds information about the impact of point mutations on the ERK2 structure–function relationship. Most of the P-ERK2 variants in the CD-site display a reduced catalytic efficiency, and for the P-ERK2 D321E, D321N, D321V and E322K, changes in thermodynamic stability are observed. The thermal stability of NP-ERK2 and P-ERK2 D321E, D321G, and E322K is decreased with respect to the wild-type. In general, a single residue mutation in the CD-site may lead to structural local changes that reflects in alterations in the global ERK2 stability and catalysis.

## 1. Introduction

Most biological processes are regulated by a complex network of protein–protein interactions (PPI) constituting critical regulatory nodes in many cell-signaling pathways [1]. The disruption of PPI due to the presence of mutation in amino acids positioned at the protein–protein interface might lead to structural loss, changes in stability, as well as functional loss or gain [2]. A detailed study on the role played by residues involved in PPIs is crucial in drug discovery and bioengineering [3] and may provide helpful information for molecule design to manipulate interaction at interfaces [4].

Protein kinases and phosphatases are involved in many signal transduction pathways mediated by specific PPIs, and a balanced state between protein kinases and phosphatases is essential for the regulation of cellular functions. Phosphorylation, on one hand, is essential for normal cellular processes, while, on the other hand, abnormal phosphorylation is one of the prime causes for alteration of structural, functional, and regulatory proteins in disease conditions, such as cancer [5,6].

The extracellular signal-regulated kinase 2 (ERK2) is a Ser/Thr kinase that belongs to the MAP kinase (MAPK) family and, with the isoform ERK1, participates to the signal transduction cascade Ras-Raf-MEK-ERK, or the ERK1/2 cascade. This cascade regulates many cellular processes, such as cell differentiation and proliferation. The ERK1/2 are activated by phosphorylation on 1 tyrosine (Tyr204/Tyr187) and 1 threonine (Thr202/Thr185) residue within the activation loop by mitogen-activated protein kinase 1 and 2 (MEK1 and MEK2) (Figure 1 and Figure 2) [7,8,9,10,11,12]. This event triggers a significant conformational change, which occurs in a cleft between the small N-terminal and the large C-terminal lobe that get closer. Both the N- and C-terminal lobes are involved in the interaction with ATP and with the protein substrate. At the N-lobe, the β-strands elements contribute to bind the adenine moiety of ATP and a conserved flexible glycine-rich loop is involved in stabilizing the interaction with ATP phosphate groups. At the N-terminal lobe, the alphaC-helix forms part of the active site by rotating and moving from the rest of the lobe. In the active conformation, the C-terminal helix flanks the alphaC helix. The activation loop, in the C-terminal lobe, is important for substrate binding and catalysis (Figure 2). The conformational changes that accompany ERK1/2 phosphorylation involve the entire protein dynamics and are essential for the proper alignment of substrates and scaffolds [13,14].

Removal of phosphate by specific phosphatases from either the tyrosine, the threonine, or from both residues inactivates the ERK2. Three groups of protein phosphatases—protein Ser/Thr phosphatase [15], protein Tyr phosphatase [16], and dual specificity phosphatases (DUSP) [17]—have been involved in the inactivation of ERKs.

The ERK2 mediates key events throughout the cell by phosphorylation of transcription factors, cytoskeletal proteins, and other protein kinases and enzymes [18,19], thus alteration of the ERK2 activity may perturb cellular homeostasis and lead to the deregulation of cellular functions. Because of its important role in signal transduction, the ERK2 is related to a variety of diseases, including cancer [20].

The ERK2 is a key regulator of the ERK pathway by recruiting interaction partners and substrates that can be either substrates or regulators through two docking sites: a conserved common docking site (CD-site), and an F-recruitment site (F-site). The CD-site is located at the opposite site of the catalytic cleft and interacts with the D-docking domain of the substrate (D-site). The F-site is in the proximity of the catalytic cleft and of the activation-loop; it is fully formed upon ERK2 activation and interacts with the F-docking domain of substrates [21,22].

The ERK2 CD-site is composed of a negatively charged component and of a hydrophobic component [21]. The negatively charged ERK2 component contains two aspartate residues (D318 and D321), essential for interaction with the positively charged basic residues of the substrate D-site [23] (Figure 2).

Several somatic ERK2 variants, identified in tumors, are reported in the *COSMIC* database [24]. Since the ERK2 plays a pivotal role in the control of different cellular processes, such as proliferation and survival, its variants may cause dysregulated cell proliferation that may induce disease, such as cancer [6,25].

One of the two positions of the ERK2 CD-site most frequently affected by mutations is D321 (Appendix A and Figure 2). This residue has been observed mutated to A, E, G, N, and V in different cancers. The substitution of D321 with N321 disrupts the binding between the ERK2 and phosphatase, thus preventing its inactivation [26]. In addition, a mutagenesis analysis revealed that substitutions of the D321 with the conservative E321 and the non-conservative A321 also rendered a modest gain of function-like phenotype [27]. Interestingly, D321E and D321A showed an increase in their phospho-active content and were also more sensitive to activation than the ERK2 wild-type [27]. Indeed, the mutations in the CD-site observed in cancer tissues are extremely interesting, first because they are present in a negatively charged ERK2 hotspot involved in binding with substrates and regulators. In addition, the substitution of a polar and negatively charged residue (D321)—with a residue ranging from a non-polar (A, V) to a neutral (N) or to a longer, negatively charged one (E)—may give the opportunity to study the role of a point mutation on ERK2 conformation in solution and stability. Interestingly, D321 is exposed to the solvent forming a salt bridge with R135, and it is closely clustered with the negative E322, which is involved in a network of interactions inside the protein [22,26]. The ERK2 E322 is another mutation hotspot and is frequently mutated to K (Appendix A). E322K represents the most frequent mutation in some cancer types [26], and increasing tumor resistance to inhibitors has been observed for this variant [28]. In a comprehensive functional study on the ERK2 missense variants, mutations in D321 and E322 belong to those ERK2 cancer-associated variants identified as gain and loss of function in cancer cells [29]. The mutations D321N and E322K change the charge in the ERK2 CD-site, may affect the ERK2 thermal stability, and may cause the evasion of dephosphorylation by dual-specificity phosphatases that leads to an increase in ERK2 activity in cells [26]. However, little is known about the consequences on the thermodynamic stability associated with the substitution of D321 and E322 in human ERK2.

In the present work, we address the impact of six non-synonymous single nucleotide variants (Appendix A) in the ERK2 CD-site on a protein structure in solution, stability, and catalytic activity. Because the activation of ERK2 by phosphorylation of T185 and Y187 induces conformational changes, we analyzed the effects of the missense mutations on the unphosphorylated (NP-ERK2) and on the phosphorylated-ERK2 (P-ERK2), obtained as pure recombinant proteins. Furthermore, we investigated the binding of CD-site mutants with inhibitors, and with peptides with kinase and phosphatase activity, respectively.

The design of new therapeutic strategies for personalized medicine requires the comprehension, from a molecular point of view, of the effect of the changes caused by single nucleotide variations on protein structure, function, stability, and interactions.

## 2. Materials and Methods

### 2.1. Site-Directed Mutagenesis

pNIC28-Bsa4 plasmid harboring the ERK2 wild-type gene was used for Escherichia coli expression. The point mutations on the wild-type ERK2 gene were introduced using QuikChange Lightning Site-Directed Mutagenesis (Agilent Technologies, Santa Clara, CA, USA). The absence of unwanted mutations was confirmed by sequence analysis, as well as the presence of the desired ones.

### 2.2. Protein Expression and Purification

The N-terminally His-tagged ERK2 wild-type and variants were expressed in *E. coli* cells Rosetta with phosphatase under appropriate conditions. The phosphatase-expressing strain has been demonstrated to be useful for the expression of protein kinases, which would otherwise auto phosphorylate. *E. coli* cells were grown in LB medium containing kanamycin (30 µg/mL final concentration) until OD at 600 nm was 0.6 AU at 37 °C. Then, 0.5 mM isopropyl-β-d-thiogalactoside (Sigma-Aldrich, St. Louis, MO, USA) was added to induce protein expression (overnight at 20 °C with shaking). The culture was centrifuged, and the pellet resuspended in 40 mL of 50 mM Hepes, 500 mM NaCl, 5 mM Imidazole, 5% glycerol pH 7.5 (Binding buffer) containing 0.5 mM Tris(2-carboxyethyl) phosphine and a cocktail of ethylenediaminetetraacetic acid (EDTA)-free protease inhibitors (Sigma-Aldrich). The cells were sonicated on ice in a Vibracell 75,115 sonicator (SONICS, Newtown, CT, USA) (3 s boots and 9 s pause), and the lysate was centrifuged. The soluble fraction was applied to a Ni-NTA (Ni2+-nitriltriacetate) affinity column (GE Healthcare, Chicago, IL, USA) pre-equilibrated with the Binding buffer. Weakly bounded contaminants were washed from the column with the Binding buffer and elution of the recombinant protein was carried out with 250 mM imidazole in the Binding buffer. The protein was then concentrated to a final volume of 2.5 mL (Amicon concentrator Ultra-15, Millipore, Burlington, MA, USA), and the imidazole was removed by a PD-10 prepacked column (GE Healthcare). Cleavage of the hexahistidine tag was carried out by overnight incubation at 4 °C with a His-tag tobacco etch virus (TEV) protease. After cleavage, the incubation mixture containing the cleaved protein, the TEV protease, and the His-tag was applied to a Ni-NTA affinity column pre-equilibrated with the Binding buffer. The flow-through containing the protein without the His-tag was collected.

Protein concentration was determined at 280 nm using a molar absorptivity of 44,810 M^−1^·cm^−1^, referred to a 41.477 kDa molecular mass. Sodium dodecyl sulfate–polyacrylamide gel electrophoresis (SDS-PAGE) was performed to check the protein purity (pre-casted NuPage 4–12% Bis-Tris polyacrylamide gels, Thermo Fisher, Waltham, MA, USA). To assess the presence of the phosphorylated (P-ERK2) and non-phosphorylated (NP-ERK2) form of the protein, western blot analyses were performed on the wild-type and the variants. The presence of the two forms was confirmed by immunodetection with the monoclonal antibody raised against the doubly phosphorylated-ERK2 (anti-Phospho-ERK1/ERK2-Thr185, Tyr187- Thermo Fisher Cat. 44–680 G-) and with the antibody anti-ERK1/ERK2 (Thermo Fisher Cat. 44–654 G), respectively (Appendix A).

For the activation of the ERK2, we used the plasmid pGEX-KG-MEKR4F (the active mutant of MEK1 kindly sent from Professor M. Cobb from Southwestern University, TX, USA) for the co-expression of the wild-type and the mutants. The ERK2 wild-type and variants were expressed as N-terminally His-tagged proteins in *E. coli* cells BL21(DE3)-pLysS under appropriate conditions. The proteins were expressed and purified, as described above.

### 2.3. Spectroscopic Measurements

Intrinsic fluorescence emission spectra (300 to 450 nm) were recorded at 20 °C in an LS50B spectrofluorometer (Perkin-Elmer, Waltham, MA,, USA), at a 100–130 μg/mL protein concentration, in 20 mM Tris/HCl, pH 7.5, 0.1 M NaCl, 200 μM DTT, in a 1.0 cm pathlength quartz cuvette.

Circular dichroism (CD) spectra in the far-UV region (190–250 nm) were recorded at a 100–130 μg/mL protein concentration (0.13 AU at 280 nm) in 20 mM Tris/HCl, pH 7.5, 0.1 M NaCl, 200 μM DTT, in a 0.1 cm pathlength quartz cuvette. Near-UV (240–420 nm) CD spectra were monitored at a protein concentration ranging over 1.0–1.3 mg/mL (1.3 AU at 280 nm) in 20 mM Tris/HCl pH 7.5, 0.1M NaCl, 1.0 mM DTT, in a 1.0 cm pathlength quartz cuvette. CD measurements were monitored in a Jasco-815 spectropolarimeter (Jasco, Easton, MD, USA), and the results were expressed as [Θ], the mean residue ellipticity, assuming a mean residue molecular mass of 110.

### 2.4. GdmCl-Induced Equilibrium Unfolding

The ERK2 wild-type and variants (75.0 μg/mL final concentration) were incubated at increasing GdmCl concentrations (0−8 M), at 4 °C, in 20 mM Tris/HCl, pH 7.5, 0.1 M NaCl, and 0.2 mM DTT. Equilibrium was reached after 30 min, then intrinsic fluorescence emission and far-UV CD spectra (0.2-cm cuvette) were recorded at 10 °C. The reversibility of the unfolding was tested by unfolding the ERK2 wild-type and variants (0.75 mg/mL final concentration) in 8.0 M GdmCl, at 10 °C, in 20 mM Tris/HCl, pH 7.5, 2 mM DTT, and 0.1 M NaCl. After 5 min, refolding was started by a 10-fold dilution of the unfolding mixture at 4 °C into solutions containing decreasing GdmCl concentrations. Equilibrium unfolding experiments were carried out in triplicate.

### 2.5. Thermal Denaturation Experiments

The ERK2 wild-type and mutants (100–130 μg/mL) were heated in a 0.1 cm quartz cuvette, from 20 °C to 90 °C, in 20 mM Tris/HCl, pH 7.5, 0.2 mM DTT, and 0.1 M NaCl. A Jasco programmable Peltier element yielded a heating rate of 1 degree × min^−1^. The dichroic activity at 222 nm and the photomultiplier were continuously monitored as described in [30]. For all thermal scans, solvent contribution at increasing temperatures was considered. Melting temperatures (T_m_) were obtained from the first derivative of the ellipticity at 222 nm, with respect to temperature [31]. All thermal transition studies were carried out in triplicate.

### 2.6. Enzyme Activity Assay and Kinetic Studies

The biochemical activity of the purified P-ERK2 wild-type and mutants was measured by monitoring the incorporation of phosphate into a peptide substrate. We used a fluorescence-based assay for rapid and sensitive detection of serine/threonine and tyrosine kinase activities, also known as the Chelation-Enhanced Fluorescence (ChEF) method. This method takes advantage of a synthetic α-amino acid with a side chain bearing an 8-hydroxyquinoline derivative (sulfonamido-oxine, Sox), which, upon coordination to Mg^2+^, gives information on the phosphorylation state of the proximal serine, threonine, or tyrosine residues in peptide- and protein-based kinase substrates. In the absence of phosphorylation, the Sox affinity for Mg^2+^ is low; upon phosphorylation, the introduced phosphate group increases the affinity of Sox fot Mg^2+^ and the fluorescence is turned on [32]. The P-ERK2 wild-type and variants activity was determined at 30 °C with the PhosphoSensR Peptide AQT0490 (AssayQuant Technologies Inc., Marlboro, MA, USA) as a substrate. The standard reaction mixture contained 50 mM Hepes pH 7.5, 10 mM MgCl2, 0.1 M DTT, 0.012% Brij-35, 1% glycerol, 0.2 mg/mL BSA, 5.0 mM MgATP, and AQT0490 ranging from 0.069 µM to 40 µM in a final volume of 0.4 mL. The reaction was started by adding different amounts of the P-ERK2 wild-type or variants, ranging from 0.008 μg to 24 μg, with a dilution obtained in 20 mM Hepes, pH 7.5, 0.01% Brij-35, 0.1 mM EGTA, 5% glycerol, 1 mM DTT, and 1 mg/mL BSA. The final enzyme concentration was 0.5 nM for D321E, 5 nM for wild-type, D321N, D321V and E322K, 10 nM for D321A and 2000 nM for D321G. The increase in the fluorescence intensity at 490 nm (excitation wavelength at 360 nm), related to the production of AQT0490 peptide phosphorylation by P-ERK2 activity, was monitored continuously for 5 min in an LS50B spectrofluorometer (Perkin-Elmer). Kinetic data were analyzed using GraphPad Prism 7.03 (La Jolla, CA, USA). Results are the mean of three experiments from different enzyme preparations.

### 2.7. Temperature Dependence of P-ERK2 Activity

The activity assay mixture (0.4 mL final volume), containing 50 mM Hepes pH 7.5, 10 mM MgCl_2_, 0.1 M DTT, 0.012% Brij-35, 1% glycerol, 0.2 mg/mL BSA, 5.0 mM MgATP, and 1 µM AQT0490 (see Section 2.6), was incubated at an increasing temperature in a thermostated cuvette. The substrate peptide concentrations used for the wild-type and variants was chosen according to the *K*_m_ values measured for each enzyme: 0.1 μM for P-D321G, and 0.5 μM for wild-type and for the other variants. The reaction was started by adding, under stirring, 2 μL of pure enzyme (kept at 10 °C) to 0.4 mL of the assay mixture equilibrated at the desired temperature (10, 15, 20, 25, 30, 35, 37, 40, 42, and 45 °C). The final enzyme concentration ranged over 0.5 nM–1400 nM. The solution was mixed in the thermostated cuvette, and the fluorescence intensity was measured at 490 nm by continuous monitoring over 5 min. Non-linear fitting to the Arrhenius equation (GraphPadPrism 7.03) of the changes of enzyme activity, as a function of temperature, was performed to obtain the activation energies (*E*a) for the catalytic reaction
*k* = *A*e^-*E*a/RT^(1)
where *k* (s^−1^) is the rate constant at temperature T (K), A is a reaction-specific quantity, R the gas constant (1.987 cal·mol^−1^·K^−1^), and *E*a is the activation energy of the reaction.

### 2.8. Data Analysis

The GdmCl-induced changes in intrinsic fluorescence emission spectra were quantified as the intensity-averaged emission wavelength, $\bar{\lambda }$, Ref. [33] calculated according to
(2)$$\bar{\lambda }=\sum (Ii/\lambda i)/\sum (Ii)$$
where λ*_i_* and I*_i_* are the emission wavelength and its corresponding fluorescence intensity at that wavelength, respectively. $\bar{\lambda }$ is an integral measurement, negligibly influenced by the noise, which reflects changes in the shape and position of the emission spectrum. GdmCl-induced equilibrium unfolding transitions monitored by far-UV CD ellipticity or intrinsic fluorescence changes were analyzed by fitting baseline and transition region data to a two-state linear extrapolation model [34], according to
Δ*G*_unfolding_ = Δ*G*^H^_2_^O^ + *m*[GdmCl] − RT ln (*K*_unfolding_)(3)
where ΔG_unfolding_ is the unfolding free-energy change for a certain GdmCl concentration, ΔG^H^_2_^O^ is the unfolding free-energy change in the absence of GdmCl, *m* is a slope term that quantifies the change in *K*_unfolding_ per unit concentration of GdmCl, R is the gas constant, T is the temperature, and *K*_unfolding_ is the equilibrium constant for unfolding. The model expresses the signal as a function of GdmCl concentration:(4)$$y_{i}=\frac{y_{N}+s_{N}{\left[X\right]}_{i}+\left(y_{U}+s_{U}{\left[X\right]}_{i}\right)\text{*}\exp\left[\left(-\Delta G{\;}^{H_{2}O}-m{\left[X\right]}_{i}\right)/RT\right]}{1+\exp\left[\frac{\left(-\Delta G^{H_{2}O}-m{\left[X\right]}_{i}\right)}{RT}\right]}$$
where *y*_i_ is the observed signal, *y*_U_ and *y*_N_ are the baseline intercepts for unfolded and native protein, *s*_U_ and *s*_N_ are the baseline slopes for the unfolded and native protein, [X]_i_ is the GdmCl concentration after the ith addition, Δ*G*^H^_2_^O^ is the extrapolated unfolding free energy change in the absence of denaturant, and *m* is the slope of a Δ*G*_unfolding_ versus [X] plot. Data were globally fitted with the *m* values shared between the data sets; all other parameters were not constrained. The denaturant concentration at the midpoint of the transition, [GdmCl]_0_._5_, according to Equation (3), is calculated as:(5)$${\left[\mathrm{GdmCl}\right]}_{0.5}=\Delta G{\;}^{H_{2}O}/m$$

GraphPad Prism 7.03 was used to fit all unfolding transition data.

Far-UV CD spectra recorded at an increasing GdmCl concentration were analyzed by a singular value decomposition algorithm (SVD) using the software MATLAB (Math-Works, South Natick, MA, USA) to remove the high frequency noise and the low frequency random errors, and to determine the number of independent components in any given set of spectra, as described in [31].

The equilibrium denaturation followed, for some ERK2 variants, a different profile, due to the formation of an intermediate at low denaturant concentration. In these cases, the changes in [Θ]_222_ or in the intrinsic fluorescence induced by increasing GdmCl concentrations were fitted to the following equation, assuming a three-state model:(6)$$\begin{array}{l} F=\frac{F_{N}+\exp\left(m_{I\text{-}N}\frac{[\mathrm{GdmCl}]-{D50}_{I\text{-}N}}{RT}\right)\cdot \left(F_{I}+F_{U}\exp\left(m_{U\text{-}I}\frac{[\mathrm{GdmCl}]-{D50}_{U\text{-}I}}{RT}\right)\right)}{1+\exp\left(m_{I\text{-}N}\frac{[\mathrm{GdmCl}]-{D50}_{I\text{-}N}}{RT}\right)\cdot \left(1+\exp\left(m_{U\text{-}I}\frac{[\mathrm{GdmCl}]-{D50}_{U\text{-}I}}{RT}\right)\right)} \end{array}$$
where *F* is λ (Equation (1)), or [Θ]_222_, *m* is a constant that corresponds to the increase in solvent-accessible surface area between the two states of the transition; *D*50_I-N_ and *m*_I-N_ are the midpoint and *m*-value for the transition between N and I, respectively, and *D*50_U-I_ and *m*_U-I_ are the midpoint and *m*-value for the transition between I and U, respectively [35]. The $\bar{\lambda }$, or the [Θ]_222,_ of the intermediate state (I), *F*_I_, is constant, whereas that of the folded state (N) and of the unfolded state (U), *F*_N_ and *F*_U_, respectively, have a linear dependence on GdmCl concentration:*F*_N_ = a_N_ + b_N_[GdmCl](7)
*F*_U_ = a_U_ + b_U_[GdmCl](8)
where a_nd_ and a_U_ are the baseline intercepts for N and U, and b_N_ and b_U_ are the baseline slopes for N and U, respectively. All unfolding transition data were fitted using GraphPad Prism 7.03.

### 2.9. Differential Scanning Fluorimetry (DSF)

The ERK2 wild-type and mutants (2 µM) in 10 mM Hepes, pH 7.5 and 0.5 M NaCl, were mixed with 12 µM inhibitors and SyPRO Orange (1:1000 dilution). After 10 min at room temperature, the increase in the fluorescence signal upon the temperature-dependent unfolding of ERK2 proteins was measured with an Mx3005p real-time PCR machine (Stratagene, San Diego, CA, USA). The ∆T_m_ shifts were calculated, as described, in [36]. The compounds that have been used were: AZD0364 (Item No. 29827), KO-947 (Item No. 29213), SCH772984 (Item No. 19166), GDC-0994 (Item No. 21107), LY3214996 (Item No. 27936), Ulixertinib (Item No. 18298), FR180204 (Item No. 15544), Magnolin (Item No. 25123) and VX-11e (Item No. 19932), purchased from Cayman.

### 2.10. Isothermal Titration Calorimetry (ITC)

Calorimetric titration experiments were carried out on VP-ITC (MicroCal) at 30 °C. The buffer used for the ERK2 wild-type and E322K was 20 mM Hepes, pH 7.5, 0.15 M NaCl, and 0.5 mM TCEP. Titration was performed by injecting the proteins (170 μM) into a reaction cell containing the inhibitor GDC 0994 (20 μM). For the measurements of both peptides, a one-time 4 µL was injected, followed by 22 injections of 8 µL with a spacing of 200 s. For the experiment with the peptide pepMEK2 and KIM of DUSP6, the ERK2 D321N (non-phosphorylated and phosphorylated) was in a 20 mM Hepes buffer, pH 7.5, 0.15 M NaCl, and 0.5 mM TCEP. For the measurements, the peptides were diluted using the same buffer to a final concentration of 500 μM. The ERK2 variant was diluted to a final concentration of 90–145 µM. The measurements were performed using an “Affinity ITC” (TA-Instruments, New Castle, DE, USA) in reversed mode, with a stir rate of 75 rpm at 30 °C. The protein was placed in the cell and the respective peptide in the syringe. For blank measurements, we used the peptides titrated in the cell containing the buffer. For the measurements of both peptides, a one time 0.5 µL was injected, followed by 44 injections of 2 µL with a spacing of 240 s. The peptides were purchased from GenScript. The sequences of pepMEK2 and the Kinase Interaction Motif (KIM) of DUSP6 peptides are MLARRKPVLPALTINP and GIMLRRLQKGNLPVRAL, respectively. They were solubilized in 5% DMSO and then diluted in buffer according to the concentrations needed for the experiment. The data were analyzed using the NanoAnalyze Data Analysis software (version 3.10.0; TA-Instruments). The corrected data were fitted to a single binding site model using a nonlinear least-square minimization algorithm, and the binding parameters, including reaction enthalpy changes (Δ*H*), reaction enthalpy changes (TΔ*S*), equilibrium dissociation constants (*K*_D_) and stoichiometry (n), were calculated. Integrated heat of the titrations, after correction for the heat of dilution, were analyzed using the Origin program.

### 2.11. Crystallization, Data Collection and Structural Determination

For crystallization experiments, a sitting-drop vapor diffusion method at 20 °C was used. The P-ERK2 variant, D321N, was a buffer exchanged into 20 mM Hepes, pH 7.5, 0.15 M NaCl, 0.5 mM TCEP, and concentrated up to 9 mg/mL. The variant was incubated with 1 mM AZD0364 (Cayman Item No. 29827) and crystallized using the reservoir solutions containing 1.6 M MgSO_4_ and 0.1 M MES, pH 6.0. D321N crystals were small needle clusters, and after 1 week were harvested. Diffraction data were collected at SLS X06SA. Data were processed with XDS [37], and scaled with aimless [38]. All structures were initially solved by molecular replacement using PHASER [39]. Manual model rebuilding was performed using COOT [40] and the structures were refined using REFMAC5 [41]. The final models were verified with MOLPROBITY [42].

### 2.12. Western Blot

Precast 4–12% polyacrylamide gels, after the electrophoretic run, were briefly rinsed with a 1× NuPAGE Transfer Buffer (Thermo Fisher), NuPAGE Antioxidant (Invitrogen), and 10% methanol [vol/vol]). Protein bands were then transferred to Hi-Bond N+ membranes (Cytiva) using a XCell II Blot Module (Thermo Fisher) at a constant voltage setting of 30 V 250 mA for 60 min. The membranes, after blotting, were washed three times with a Tris-buffered saline (TBS) with 0.1% Tween^®^ 20 detergent (TBST) and blocked with TBST and 5% Bovine Serum Albumin (BSA, Sigma Aldrich). The membranes, after blocking, were washed once with TBS-T, and twice with TBS before the incubation with the primary Antibody 44–680 G anti-Phospho-ERK1/ERK2 (T185, T187) (Thermo Fisher, Catalog# 44–680 G) monoclonal antibody raised against the doubly phosphorylated-ERK2. The 1:5000 diluted primary antibody incubation was carried out overnight at 25 °C. The membranes, with the bound primary antibody, were incubated with an anti-Rabbit IgG (Thermo Fisher, Catalog# 32460) and diluted 1:5.000 in TBS for 60 min at 25 °C, and then washed three times in TBST. Labeled protein bands were detected using a chemiluminescent system, according to the supplier protocol (Amersham). The same western blot membranes were used to re-probe with the antibody raised against the non-phosphorilated form of the ERK2 Antibody anti ERK1/ERK2 (Thermo Fisher, Catalog# 44–654 G) after the application of the following stripping protocol: the blots were incubated in a 0.1 M Glycine-HCl buffer pH 3.0 for 30 min at 25 °C with agitation, then were washed once with distilled water and twice with a TBS buffer before proceeding with the blocking solution (TBST + BSA 5%) for 60 min at 25 °C. The membranes proceeded to the next round of immunodetection with the anti ERK1/ERK2 antibody following the previously described protocol.

## 3. Results and Discussion

To study the consequences of a single amino acid substitution on an extracellular signal-regulated kinase 2 (ERK2) common docking site (CD-site), we characterized six missense ERK2 mutations found in cancer tissues (D321A, D321E, D321G, D321N, D321V and E322K) and reported in COSMIC database [24] (Appendix A). The two hotspot variants, D321N and E322K—found with a higher frequency in cancer tissues—caused an increase in ERK2 cellular activity and evaded inactivation by dephosphorylation [26]. The ERK2 activity is regulated by the phosphorylation of T185 and Y187 by mitogen-activated protein kinase 1 and 2 (MEK1 and MEK2) (Figure 1B and Figure 2).

All the variants’ object of this study were obtained as recombinant proteins using site-directed mutagenesis and resulted as soluble pure proteins in the non-phosphorylated-ERK2 (NP-ERK2) and phosphorylated-ERK2 (P-ERK2) conformations (see Materials and Methods). The biophysical characterization was carried out on both NP-ERK2 and P-ERK2 wild-types and variants. Phosphorylation was checked by western blot analysis (Appendix A).

The conformation in the solution of NP-ERK2 and P-ERK2 wild-types and mutants was analyzed by circular dichroism (CD) and fluorescence spectroscopy. The near-UV CD spectra of the NP-ERK2 and P-ERK2 wild-type showed a positive spectral contribution at around 290 nm, typical of tryptophan residues, with fine structure features at 260–275 nm (Figure 3A). The two spectra showed similar contributions in the tryptophan region, while in the region between 250 and 280 nm, the contributions of phenylalanine and tyrosine residues were less pronounced for the P-ERK2, suggesting a conformational change in the protein upon phosphorylation.

The fluorescence spectra of the NP- and P-ERK2 wild-types (Figure 3B) showed the same maximum emission wavelength at around 339 nm, with differences in the relative fluorescence emission intensity, which is slightly increased in the case of the phosphorylated form. Far-UV CD spectra of the NP-ERK2 and P-ERK2 wild-types showed a zero intercept at around 200 nm and 2 minima at around 208 and 222 nm, suggesting an alpha helical major contribution, slightly influenced by β sheets elements (Figure 3C).

The spectra of NP-ERK2 variants in the near-UV CD region were similar to that of the NP-ERK2 wild-type, except for D321G (Figure 4A). In particular, the near-UV CD spectrum of D321G lacks the contribution of the tryptophan residue and a complete inversion of the dichroic activity was observed in the region between 260 and 275 nm. Interestingly, the substitution of D321 with the small and more flexible glycine perturbs the network of ionic interactions between D321 and R135 and may alter the ERK2 tertiary arrangement. The near-UV CD spectra of most P-ERK2 variants (Figure 4B) were comparable, but not identical, with that of the P-ERK2 wild-type, with two notable exceptions: P-ERK2 D321G, which showed a near-UV CD spectrum like that of NP-ERK2 D321G; and P-ERK2 E322K, whose near-UV CD spectrum did not show a well-defined contribution in the 260–275 nm region. Amongst the other P-ERK2 mutants, only the P-D321V spectrum in the near-UV CD region was like that of the P-ERK2 wild-type. These results, taken together, indicate that, upon phosphorylation, changes occur in the tertiary arrangements of the proteins and that the residues in the CD-site may be involved in these interactions.

The fluorescence spectra of the P-ERK2 D321E variant was centered at 343 nm, and it was significantly different from that of the wild-type (Figure 4D), suggesting changes in the tryptophan environment. For the NP-ERK2 variants, only D321N showed a slight increase in the relative fluorescence intensity (Figure 4C); no significant differences were detected for all the other variants.

The analysis of far-UV CD spectra revealed that the secondary structure of the NP-ERK2 variants was similar to that of the wild-type (Appendix A), hence the effect of the single amino acid substitution was mainly localized to the structural environment of the mutated residue. The exception was the NP-ERK2 D321G, which showed a difference in the intensity and in the minima around 208 and 222 nm, in comparison with that of the wild-type, supporting the observation that aspartate-to-glycine substitution at residue 321 significantly affects the ERK2 native structure. The far-UV CD spectra of the P-ERK2 wild-type and variants were similar, with a divergence in the intensities of the signals, suggesting a different secondary structure rearrangement of the mutants upon phosphorylation (Appendix A).

Far-UV CD is an excellent method to study the impact of the point mutation on the secondary structure, and the ratio between molar ellipticity at 222 and at 208 nm ([Θ]_222_/[Θ]_208_) is very sensitive to structural changes. This parameter, useful to distinguish between coiled coil helices (>1.0) and non-interacting helices (0.8–0.9), may give information about interhelical contacts present in a helix bundle and coiled coil structures [43]. The [Θ]_222_/[Θ]_208_ for the wild-type is 0.8 and ranges from 0.76 for NP-D321N to 0.98 for NP-D321G. This variation suggests that the single amino acid substitutions in the CD-site can alter interhelical interactions in the solution (Appendix A).

A missense mutation may lead to significant structural alterations that may change the stability of the protein and impair the protein function [44,45]. To provide insights into the effect of missense mutations on protein stability, we analyzed the thermal stability of the NP-ERK2 and P-ERK2 wild-types and variants by continuously monitoring the changes of the molar ellipticity at 222 nm in the temperature range between 20 °C and 90 °C (Figure 5 and Appendix A). The irreversible thermal denaturation occurred in a transition apparently cooperative. The melting temperature (T_m_), corresponding to the midpoint of the denaturation process, was calculated by plotting the first derivative of the molar ellipticity values as a function of temperature. The thermal transition of the NP-ERK2 wild-type (T_m_ = 55.0 °C) was less cooperative than that of the P-ERK2 (T_m_ = 56.0 °C) (Figure 5).

In general, upon phosphorylation, the melting temperature increases for the P-wild-type, P-D321A, P-D321E and P-D321G, and decreases for P-D321N, P-D321V, and P-E322K (Table 1).

T_m_ values were increased for the NP- and P-D321N (Table 1), whose T_m_ values were 4 and 2 degrees higher than that of the wild-type, respectively. For the NP-ERK2 D321V, the T_m_ value is 2 degrees higher than the NP-ERK2 wild-type, and in the case of the P-ERK2 D321V, the T_m_ value was about 3 degrees lower than that of the NP-ERK2 D321V, and slightly lower than the T_m_ of the P-ERK2 wild-type. The T_m_ values of the NP-ERK2 and P-ERK2 D321E, D321G, and E322K variants were lower than those of the wild-type: T_m_ values of the NP-E322K and P-E322K were 4 degrees and 6 degrees lower than that of the NP-ERK2 and P-ERK2 wild-type, respectively (Table 1). These results indicated that the negatively charged residues in the CD-site were involved in the stability of the native state and that a charge inversion in E322 may result in a dramatic thermal stability loss. Notably, E322 points to the internal region of the protein where it is involved in buried ionic interactions that may be disrupted by the substitution of the negatively charged glutamate with the positive lysine. Additionally, the residue D321, involved in a network of ionic interactions with R135, is critical for ERK2 stability since a substitution of the negatively charged aspartate with a positively charged residue—as in D321N—or with a neutral—as in D321A and D321V—or with the longer glutamate—as in D321E—affect the protein thermal stability. In D321G, the presence of a neutral, small, and flexible glycine that significantly alters the protein tertiary arrangement (Figure 4) brings about a decrease in its thermal stability (Table 1). The impact of mutation on the negatively charged residues in the CD-site, D321 and E322, on the ERK2 thermal stability was also observed in rat ERK2 variants, D321N and E322K, and points to the significant role of the CD-site as an energetic hot spot in this protein [26].

The thermodynamic stability of the ERK2 wild-type and mutants was analyzed at 10° C using GdmCl as a denaturant, by monitoring, in parallel, the changes in the intrinsic fluorescence and in the molar ellipticity at 222 nm. The changes in the intrinsic fluorescence upon increasing [GdmCl] were measured by calculating $\bar{\lambda }$, the intensity averaged emission wavelength (Equation (2)). The Appendix A reports the thermodynamic parameters resulting from the analysis of the changes in the far-UV CD and in the intrinsic fluorescence caused by the increasing GdmCl concentration. The GdmCl induced unfolding transitions monitored by far-UV CD changes are coincident for most of the variants with those obtained from changes in intrinsic fluorescence. For some variants, the unfolding transition monitored by far-UV CD and/or by fluorescence was analyzed using a three-state model since an intermediate was detected at a low denaturant concentration (Appendix A). Notably, all the variants showed conformational stability that, in general, were very similar to that of the wild-type, with some notable differences as in the case of D321E, D321N, D321V, and E322K. In these variants, important changes occur upon phosphorylation in the thermodynamic parameters and a denaturation intermediate becomes populated. The presence of a denaturation intermediate could be regarded as a stabilization of the native state, because of the increased difference in the unfolding free energy between the native and the unfolded state. The general consideration about these results is that a single residue mutation in the CD-site reflects in important alterations in the global ERK2 thermodynamic stability. In the case of D321G, the denaturation intermediate is populated in the unphosphorylated form, and the stability of the phosphorylated form is comparable to that of the wild-type.

The effect of missense mutations on the kinetic properties of ERK2 was investigated by monitoring over time the increase in the fluorescence intensity induced by the incorporation of phosphate into a peptide substrate (PhosphoSens^R^ Peptide AQT0490). The activity assay was performed at 30 °C at a fixed MgCl_2_ concentration.

Most of the P-ERK2 variants in the CD-site displayed a reduced catalytic efficiency (*k*_cat_/*K*_M_), analyzed using AQT0490 over the concentration range of 0.069 µM to 40 µM (Table 2). P-D321E and P-D321A displayed a 19- and 3-fold increase in the specific activity, respectively, not paralleled by a similar increase in their catalytic efficiency (*k*_cat_/*K*_m_), that is, a 1.5–fold decrease for P-D321A and only a 3.8-fold increase for P-D321E (Table 2). This result—that originates from the increase in *K*_m_ values of about four-fold (Table 2) for both the variants—suggests a decrease in substrate binding. A significant decrease in the specific activity was observed for the variant P-D321G, as indicated by the low *k*_cat_/*K*_M_ value and specific activity (Table 2). Notably, the tertiary arrangement of this variant is significantly altered (Figure 4) as a result of the substitution of D321 with the small and more flexible glycine that may perturb the network of ionic interactions between D321 and R135.

The P-ERK2 variants in the CD-site displayed an optimal temperature for catalysis (T_max_), higher than that of the wild-type, except for the variant P-D321V that showed a T_max_ at 25 °C, suggesting that in physiological conditions, this variant may be completely inactive (Table 3 and Appendix A). Notably, the variant with the highest specific activity, P-D321E, showed the lowest *E*_a_ value (5.24 kcal/mol) (Table 3), comparable to the activation energy value of P-D321N. The *E*_a_ values of all the other mutants are similar to that of the wild-type.

The ERK2 CD-site is in a region where the major changes between inactive (NP-ERK2) and active (P-ERK2) proteins are observed. In fact, when the CD-site binds to the activating and deactivating enzymes, solvent exposure of the activation loop occurs, thus suggesting a relation between the activation loop and the CD-site [46].

To correlate structural and stability studies, we focused on the effects of the variants in the binding with inhibitors, in particular the inhibitors that occupy part of the adenine binding pocket and form hydrogen bonds with the hinge region of the enzyme, known as type I inhibitors of ERK2 (Appendix A) [47,48,49,50,51,52,53,54]. The binding to the inhibitors was studied by Differential Scanning Fluorimetry (DSF) measurements that revealed, for all the variants, T_m_ values like that of the wild-type, with a small difference of 2–4 °C (Appendix A), suggesting that the missense mutations do not affect the binding to ERK2 inhibitors.

The hotspot mutation E322K is associated to an apparent increase in tumor resistance to inhibitors [26,30]. Isothermal titration calorimetry (ITC) assay was performed to quantify the effect of this mutation on the binding with a specific inhibitor, GDC-0994 [52]. ITC is a convenient and widely used approach to directly measure the amount of heat released or absorbed during association processes of biomolecules in solutions and to quantitatively estimate the interaction affinity between protein and its ligands. Titration of the ERK2 wild-type with the inhibitor GDC-0994 showed an affinity (*K*_D_) of about 3.8 nM—in the same order of magnitude of that of the variant E322K (*K*_D_ = 7.9 nM). Thermodynamic measurements provide insight into the nature of the noncovalent forces responsible for binding; polar interactions tend to contribute favorably to the enthalpic component, whereas entropically favored interactions are often hydrophobic and are dominated by dehydration. The binding of the ERK2 wild-type and the inhibitor GDC-0994 revealed an enthalpy-driven interaction, driven by favorable polar interactions such as hydrogen-bonding. On the other hand, the interaction of ERK2 E322K and the inhibitor GDC-0994 displayed an entropy-driven interaction, mainly characterized by contribution of hydrophobic effects to binding.

To better understand the effect of the D321N mutation on the interaction with the protein kinase MEK2 and the dual specificity phosphatase (DUSP6), we performed ITC titrations with the peptides PepMEK2 and the Kinase Interaction Motif (KIM) of DUSP6. PepMEK2 is a MAP Kinase ERK2 substrate docking peptide derived from MLARRKPVLPALTINP of the upstream activating kinase MEK2. The binding with the pepMEK2 of the ERK2 wild-type was analyzed and compared with that of the variant D321N. ITC data revealed an enthalpy-driven interaction of the NP-ERK2 wild-type with pepMEK2, with a *K*_d_ value of 23.7 µM. The titration profile of the pepMEK2 with the NP-ERK2 D321N displayed an entropy-driven interaction with a *K*_d_ value of 11.6 µM. These results indicate that the amino acid substitution in D321N perturbs the docking interaction in the CD-site.

The D321N variant shows an increased cellular activity in cells and evades inactivation catalyzed by phosphatases, such as DUSP6 [26]. DUSP6 plays a crucial tumor-suppressive role via pivotal negative-feedback regulation of the ERK2 in the MAPK signaling pathway [55,56]. The KIM peptide GIMLRRLQKGNLPVRAL of DUSP6 directly binds to the ERK2 CD-site. The ITC titration between the P-ERK2 wild-type and the KIM peptide DUSP6 showed a *K*_d_ of 14.4 µM (Figure 6). This analysis shows an enthalpy-driven interaction between hydrogen-bonding and van der Waals interactions. The P-ERK2 D321N showed no detectable binding heat with the KIM of DUSP6, suggesting that the loss of the aspartate negative charge in the asparagine variant prevents any binding interaction with KIM [26].

To provide a structural model for future inhibitor development, we crystallized the phosphorylated-ERK2 mutant D321N with AZD0364 (Cayman Item No. 29827) (Figure 7). Surprisingly, the activation segment was partially disordered, possibly caused by inhibitor binding.

Apart from this difference, the structure was highly conserved compared with the wild-type ERK2 (Figure 8).

Unanticipated was also the presence of the inhibitor that interacted not only with the ERK2 binding pocket for ATP, but also bound to the crystal interface (Figure 8). As this binding site is formed by crystal contacts, it is likely an artefact of protein crystallization.

The studies on purified ERK2 variants may contribute to understand the effect of missense mutations on protein function and structure [57]. The results drive to conclude that the effect of mutations in the CD-site is limited to local changes of tertiary structure, whereas the global fold of the ERK2 is not drastically affected, as well as the binding of inhibitors. Indeed, the local changes strongly reflect on the global properties of the ERK2 as the thermodynamic parameters or the activation energy values. Notably, the conformational stability of all the variants was comparable, but not identical, to that of the wild-type, indicating that a single residue mutation in the energetic hotspot of the CD-site brings about changes that globally reflect on the ERK2 stability. The changes in the enzyme activity observed for some of the ERK2 variants suggest a long-distance effect at the active site produced by the mutation in the CD-site that is located far away from the catalytic site. Mutations in the ERK2 CD-site residues, a region where the major changes between inactive (NP-ERK2) and active (P-ERK2) proteins are observed, may affect the transition between active and inactive conformations, as suggested by the absence of binding interaction of the P-ERK2 D321N with the KIM peptide of DUSP6.

## 4. Conclusions

The missense variants in the ERK2 common docking site (CD-site), a region involved in interactions with substrates and regulators, are considered statistically significant in cancer. Biophysical studies on these variants present in cancer tissues indicate that the ERK2 CD-site, that is far away from the catalytic site, plays a significant role in the energetics of the protein. In this site, the major changes between inactive (NP-ERK2) and active (P-ERK2) proteins are observed, thus amino acid substitutions in this region may affect the transition between the active and the inactive conformation and alter binding interactions with regulators, such as phosphatase DUSP6. The lack of control of the ERK2 activity by DUSP6 may keep the ERK2 signaling active, thus favoring oncogenic transformation. A single residue mutation in the CD-site brings about local changes that strongly reflect on the global properties of the ERK2, as the thermodynamic parameters or the activation energy values significantly changed in some of the mutants. The negatively charged residues in the CD-site are involved in the stabilization of the native state and a charge inversion or neutralization of residues D321 or E322 may result in a significant change in the ERK2 biophysical and biochemical properties. The differences observed in the catalytic parameters, as the reduced catalytic efficiency, may result from a different response to the control mechanism that regulates the transition from active- to inactive-conformation of the overall protein structure.

## Figures and Tables

**Figure 1 cancers-15-02938-f001:**
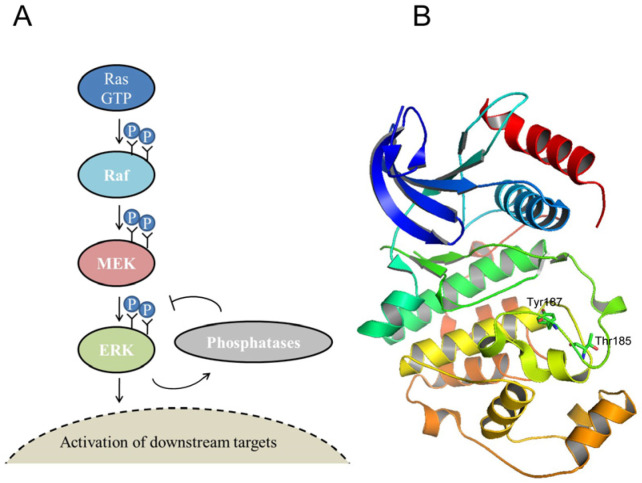
The ERK1/2 cascade and ERK2 structure. (**A**) ERK1/2 signaling cascade. The ERK kinases from the cytoplasm can be translocated into the nucleus and catalyze the phosphorylation of downstream targets upon activation. Mitogen-activated protein kinase 1 and 2 (MEK1 and MEK2) activate the ERK2 by phosphorylation of two residues, Thr185 and Tyr187. (**B**) Human ERK2 structure in complex with an inhibitor (pdb: 4zzn) [12]. Thr185 and Tyr187 residues, involved in the activation of ERK2, are evidenced in sticks; the inhibitor is not shown.

**Figure 2 cancers-15-02938-f002:**
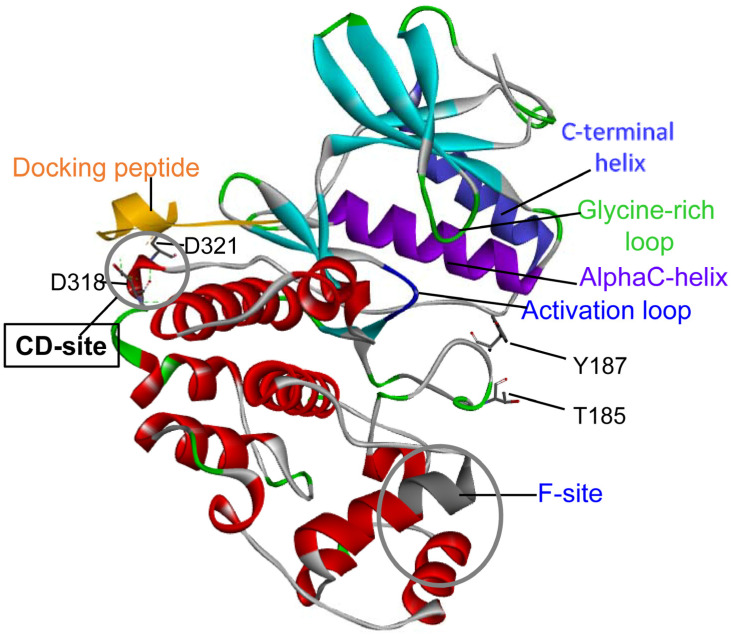
Model of the human non-phosphorylated-ERK2 (NP-ERK2) CD-site. The CD-site is the common docking site that contains residues that interact with protein substrates and is composed of two negatively charged residues, D318 and D321, depicted in sticks. In orange, the docking peptide derived from hematopoietic tyrosine phosphatase, a negative regulator of ERK2, sequence in a one letter code: RLQERRGSNVALMLDC. In the NP-ERK2 structure (pdb: 2gph), the docking peptide is involved in extensive electrostatic interactions with the CD-site. All the other relevant parts of the ERK2 structure are shown: in dark blue and in light green, the activation loop and the glycine-rich loop, respectively; in purple and in light blue, the helices alphaC and C-terminal, respectively; in the grey circle, the F-site. The T185 and Y187 at the active site are represented in sticks.

**Figure 3 cancers-15-02938-f003:**
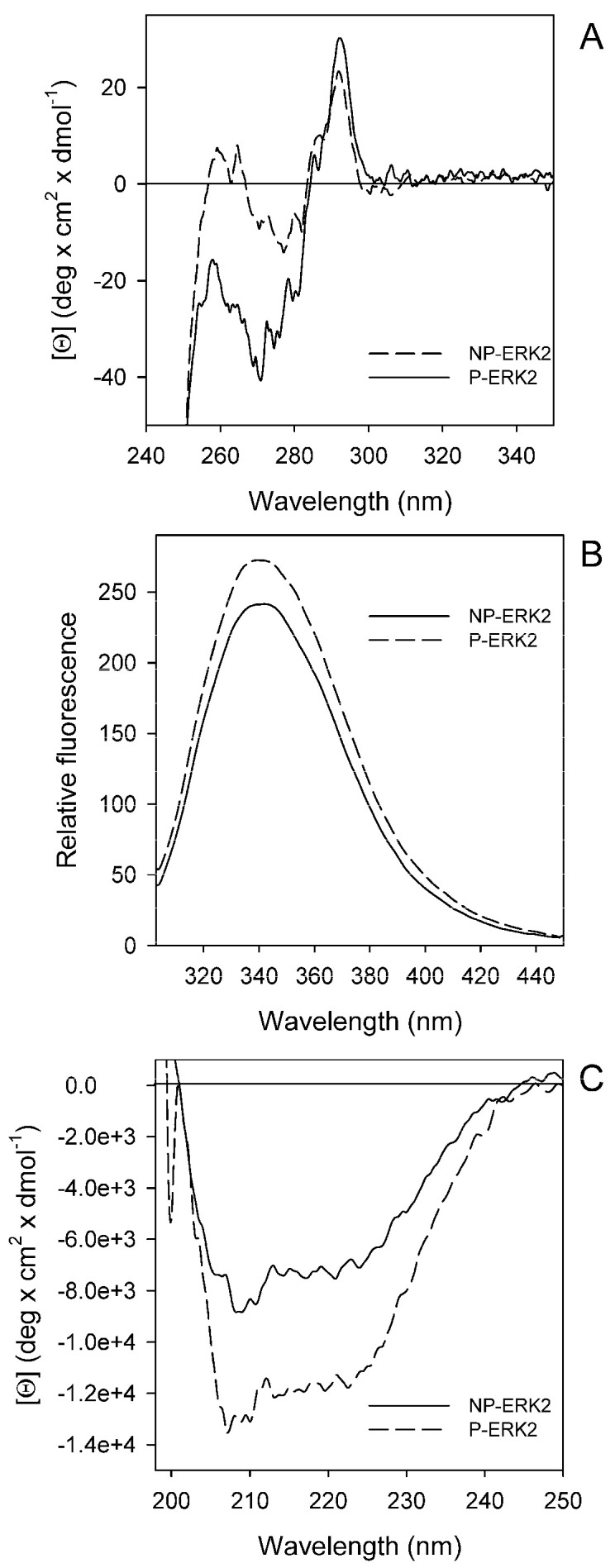
Spectral properties of the non-phosphorylated-ERK2 (NP-ERK2) and phosphorilated-ERK2 (P-ERK2) wild-types. (**A**) Near-UV CD spectra were recorded in a 1.0-cm quartz cuvette at 1.3 mg/mL protein concentration in 20 mM Tris-HCl, pH 7.5, containing 1.0 mM DTT and 0.1 M NaCl. (**B**) Intrinsic fluorescence emission spectra (295 nm excitation wavelength) were monitored at 130 µg/mL (0.08 AU 280 nm) in 20 mM Tris-HCl, pH 7.5, 0.1 M NaCl, and 0.2 mM DTT. (**C**) Far-UV CD spectra were monitored in a 0.1-cm quartz cuvette at 130–170 µg/mL in 20 mM Tris-HCl, pH 7.5, 0.2 M NaCl, and 0.2 mM DTT. The continuous lines are used for the NP-ERK2, the dashed lines are used for the P-ERK2. All spectra were recorded at 20 °C.

**Figure 4 cancers-15-02938-f004:**
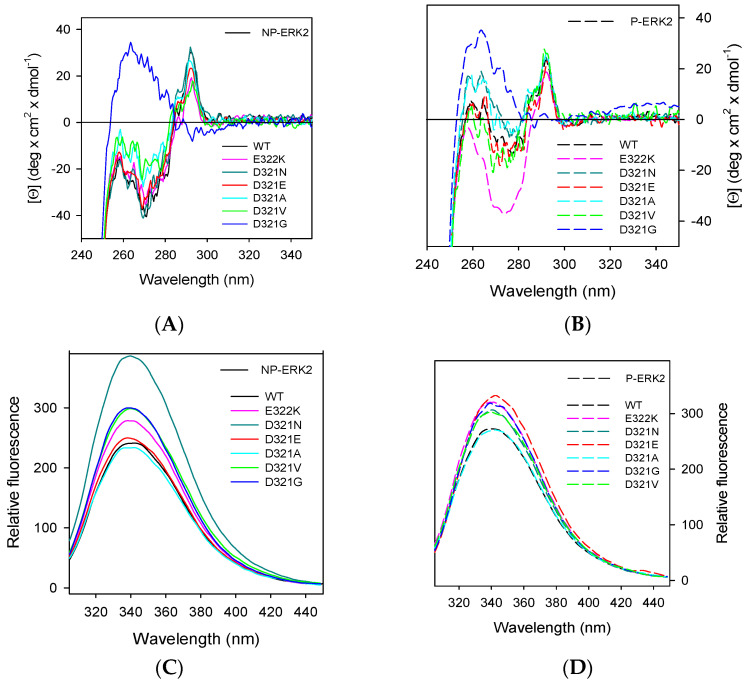
Spectral properties of the non-phosphorylated-ERK2 (NP-ERK2) and phosphorylated-ERK2 (P-ERK2) variants in the CD-site. Near-UV CD spectra of the NP-ERK2, continuous line (**A**), and of P-ERK2 variants, medium-dashed line (**B**), were monitored at 1.3 mg/mL protein concentration in 20 mM Tris-HCl pH 7.5, 1.0 mM DTT, and 0.1M NaCl, in a 1.0-cm quartz cuvette. Intrinsic fluorescence emission spectra of the NP-ERK2, continuous line (**C**), and of the P-ERK2 variants, dashed lines (**D**), were recorded at 130 μg/mL (0.08 AU 280 nm, 295 nm excitation wavelength), in 20 mM Tris-HCl, pH 7.5, 0.1 M NaCl, and 0.2 mM DTT.

**Figure 5 cancers-15-02938-f005:**
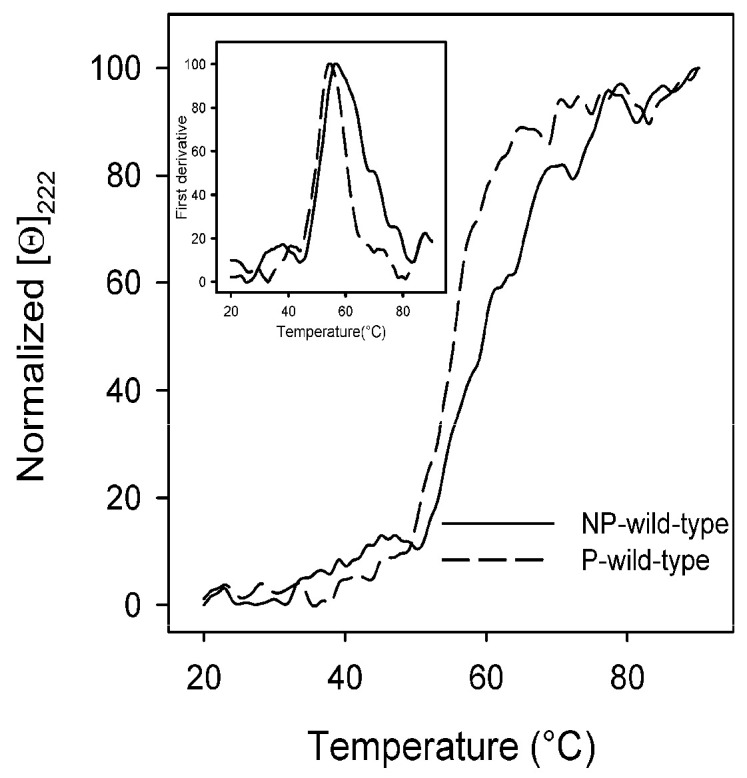
Thermal unfolding of the non-phosphorylated-ERK2 (NP-wild-type) and phosphorylated-ERK2 (P-wild-type) wild-types. The ERK2 wild-type proteins (100–130 μg/mL), in 20 mM Tris-HCl, pH 7.5, 0.1 M NaCl, and 200 µM DTT, were heated from 20 °C to 90 °C. The molar ellipticity at 222 nm ([Θ]_222_) was monitored continuously every 0.5 °C and normalized. The first derivative of the thermal transition data is shown in the inset.

**Figure 6 cancers-15-02938-f006:**
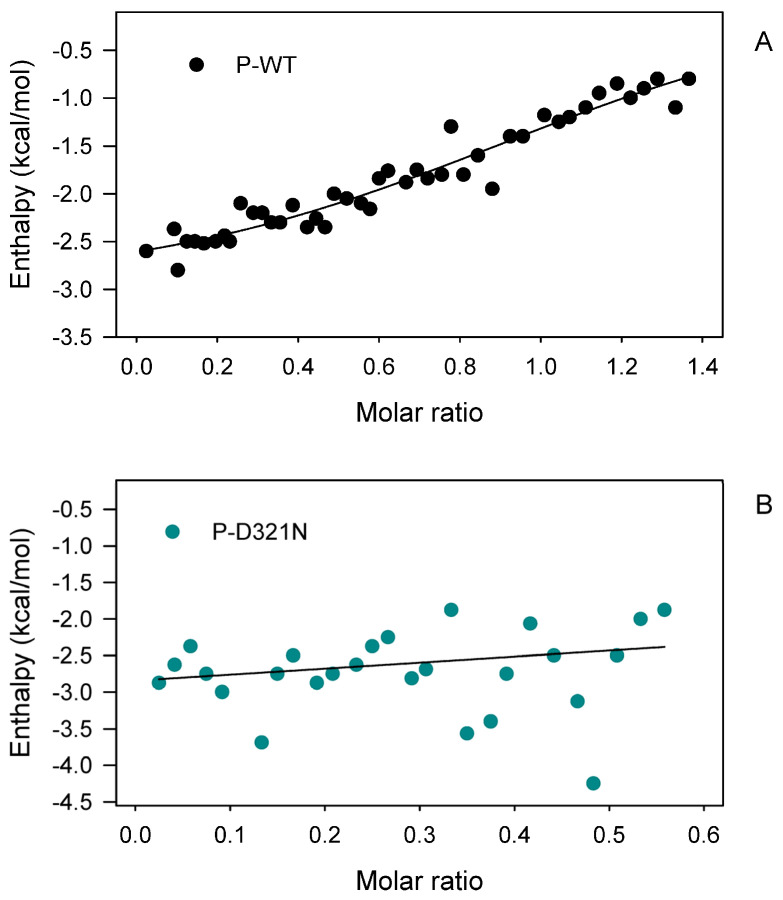
Effect of the binding with KIM peptide of DUSP6 on the amount of heat released or absorbed during association of the phosphorylated-ERK2 (P-ERK2) wild-type (**A**), and the P-ERK2 D321N variant (**B**). The measurements were performed using an “Affinity ITC” (TA-Instrument) with 500µM of peptide (KIM of DUSP6) in the syringe, and 93 µM of the P-ERK2 wild-type and 113 µM of the P-ERK2 D321N in the cell at 30 °C. The K_D_ was 14.4 µΜ for the phosphorylated wild-type (**A**), no binding was measured for the phosphorylated D321N (**B**).

**Figure 7 cancers-15-02938-f007:**
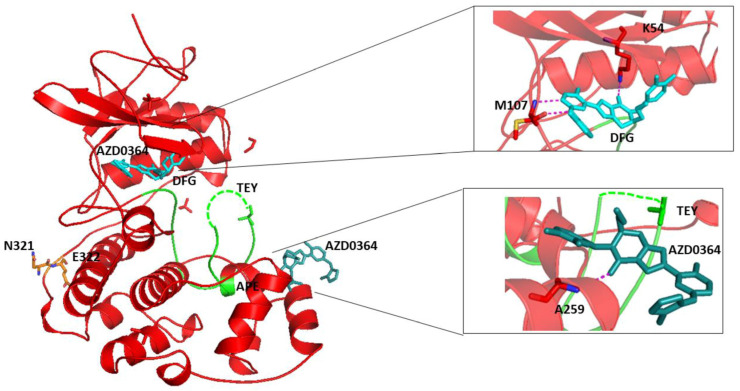
The human phosphorylated-ERK2 (P-ERK2) D321N (in red) bound to inhibitor AZD0364 (in cyan). In green, the activation loop that begins with the sequence DFG and ends with the sequence APE. The dotted line underlines the undefined region of the three TEY amino acids, important for the activation of ERK2. In cyan, the compound AZD0364 that binds in two opposite regions the mutant D321N.

**Figure 8 cancers-15-02938-f008:**
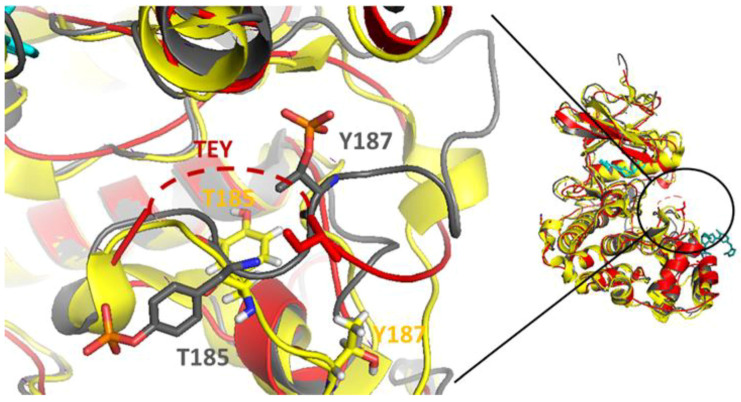
Structural comparison of the human phosphorylated-ERK2 (P-ERK2) D321N bound to inhibitor AZD0364 (in cyan). Superposition of the human P-ERK2 D321N (in red), rat P-ERK2 wild-type (in yellow, pdb: 2erk), and rat NP-ERK2 D321N (in gray, pdb: 6ot6). The residues T185 and Y187, important for the activation of ERK2, are depicted in sticks. The dotted line underlines the undefined region of the three TEY amino acids involved in ERK2 activation.

**Table 1 cancers-15-02938-t001:** Melting temperatures for the phosphorylated-ERK2 (P-ERK2) and non-phosphorylated-ERK2 (NP-ERK2) wild-types and variants in the CD-site.

	T_m_ (°C)	
ERK2	NP-ERK2	P-ERK2
wild-type	55.0	56.0
D321A	56.0	57.1
D321E	54.0	55.1
D321G	54.0	55.0
D321N	59.0	58.1
D321V	57.0	54.5
E322K	51.0	50.3

T_m_ values were obtained from the first derivative of the ellipticity at 222 nm at an increasing temperature.

**Table 2 cancers-15-02938-t002:** Kinetic parameters of the phosphorylated-ERK2 (P-ERK2) wild-type and variants in the CD-site.

P-ERK2	*K*_M_ (μM)	*k*_cat_ (s^−1^)	*k*_cat_/*K*_M_ (s^−1^·μM^−1^)	Vmax (μM·s^−1^)	Specific Activity (μM·s^−1^·μg^−1^)
wild-type	1.81 ± 0.29	2.20	1.215	1.101 × 10^−2^	1.218
D321A	7.42 ± 2.69	5.93	0.799	5.925 × 10^−2^	3.571
D321E	7.11 ±1.77	32.96	4.635	1.648 × 10^−2^	19.867
D321G	0.65 ± 0.12	1.64 × 10^−4^	2.523 × 10^−3^	3.278 × 10^−3^	0.001
D321N	2.95 ± 0.76	1.26	0.427	6.307 × 10^−3^	0.760
D321V	3.47 ± 0.55	2.41	0.694	1.207 × 10^−2^	1.455
E322K	4.70 ± 0.81	1.64	0.349	8.223 × 10^−3^	0.099

The catalytic activity of the P-ERK2 was assayed at 30 °C, with the substrate peptide AQT0490 (0.1 μM for P-D321G, and 0.5 μM for wild-type and for the other variants) in the presence of 5 mM MgATP (see Materials and Methods). The final enzyme concentration in the activity assay was 0.5 nM for P-D321E, 5 nM for wild-type, P-D321N, P-D321V and P-E322K, 10 nM for P-D321A, and 2000 nM for P-D321G. The ERK2 activity kinetic parameters were measured at 30 °C with at least 10 different AQT0490 concentrations. Data are the mean ± SE of the fit.

**Table 3 cancers-15-02938-t003:** Temperature effect on kinase activity of the phosphorylated-ERK2 (P-ERK2) wild-type and mutants in the CD-site.

P-ERK2	T_max_ (°C)	*E*_a_ (kcal/mol)
wild-type	30.0	9.95 ± 0.41
D321A	35.0	11.43 ± 1.40
D321E	40.0	5.24 ± 0.39
D321G	35.0	8.97 ± 0.50
D321N	35.0	6.67 ± 0.49
D321V	25.0	10.12 ± 1.49
E322K	40.0	9.76 ± 1.58

*E*a was determined in the temperature range between 10 °C and the optimal temperature (Equation (1)). Data are the mean ± SE of the fit.

## Data Availability

The data presented in this study are in the article or in Appendix A.

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
