# Peer review of "Mutation in the Common Docking Domain Affects MAP Kinase ERK2 Catalysis and Stability"

_cancers, 2023, doi:10.3390/cancers15112938_

Round 1

Reviewer 1 Report

no comments

Author Response

Novak et al. investigated biophysical and biochemical properties of Erk2 molecules, mutated at Asp321 and E322, critical residues of the so called common docking domain. The Erk MAP kinases are pivotal signaling molecules, often unregulated in human diseases, particularly in cancers. Their abnormal activity in tumors is commonly a consequence of activating mutations in upstream components (RTKs, Ras, Raf…). Mutations are rarely, monitored in ERKs too, and among those the mutation E322K is the most frequent. Mutations in D321 are also found. These mutations have been studied quite intensively in vitro, in flies and in mammalian cells, and it is already known that they do not affect intrinsic catalytic properties, but rather binding affinities, primarily to phosphatases. The current work adds some information with respect to the spectral properties of the mutants (in their phosphorylated and non-phosphorylated states), their thermal stability, kinetic parameters of the activity and binding to peptide (derived from the phosphatase DUSP6). The author also looked at the crystal structure of Erk2(D321N) bound to a pharmacological inhibitor.

The study is informative, but the novel knowledge obtained is marginal. In most assays the differences between native Erk2 and mutants are not dramatic. An exception is perhaps the lack of binding to the DUSP peptide, but the reduced affinity to phosphatases is not novel. In the cases in which there is an unusual property of a mutant, it is restricted to one mutation with no idea why this given mutation imposed the difference (for example the spectral properties of ERK2(D321G). It should be an editorial decision whether a study that provides results which are in a way important (given the importance of Erk kinases), but mostly negative, merit publication.

In any case the authors are advised to improve the presentation of the paper. The abstract and the conclusion chapter are vague and general.

We thank the Reviewer for the careful reading of the manuscript and for all the comments that helped in improving the quality of the manuscript.

Point 1: The abstract does not really represent the content and shares no explicit results.

Response 1: the Abstract has been modified according to the referee’s suggestion.

Point 2: The conclusions seem similarly not associated the study; Strange.

Response 2: This section has been rewritten, according to the referee’s suggestion.

Point 3: The Introduction is also narrow, missing information most relevant to the study. It ignores the ERK pathway, does not mention even ERK1 and the fact that mutations found in it seem oncogenic, capable of transforming cells (unlike the mutations in ERK2, including mutations studied here). It must include a deeper description of Erk’s structure-function relationships, and description of the two substrate binding domains, not only the CD.

Response 3: the introduction has been modified accordingly.

Point 4: Finally, there is really no need for 3 repeated figures of the Erk2 crystal structure. Figure 3 could be removed altogether.

Response 4: Figure 3 has been removed, as suggested.

Point 5: The result section is generally fluent, but this reviewer was somewhat confused by Table 2:

- What is the meaning of ‘Enzyme concentration’ column? Isn’t the concentration determined by the experimenter? I believe this parameter has to do with the particular assay applied here – in any case, it must be better explained (I did look at the M&M).

Response 5: The “Enzyme concentration” column reports the final enzyme concentration used in the activity assay to have a similar substrate consumption for all the variants, so to guarantee steady-state conditions. In particular, for P-D321E, whose catalytic efficiency (kcat/KM) and specific activity were higher than those of the wild-type, we used ten-fold less enzyme. In the case of P-D321G, whose catalytic efficiency (kcat/KM) and specific activity were significantly lower than those of the wild-type, a 400-fold higher concentration of enzyme was necessary. We removed this column from Table 2 in the revised version because we agree with the referee’s observation that it was confusing for the reader. The information has been added as a footnote to Table 2 and in the sections 2.6 and  2.7.

- Same question rises for the column of substrate (AQT0490) concentration.

Response: The column of substrate (AQT0490) concentration refers to the concentration used in the enzyme assays to determine the specific activity (Table 2) or the activation energy (Table 3). In both these measurements the substrate (AQT0490) concentration was well below the Km value to prevent the formation of the ES complex and thus measuring only the product release. We removed this column from Table 2, in the revised version, because we agree with the referee’s observation that it was confusing for the reader. The information has been added as a footnote to Table 2 and in the sections 2.6 and  2.7.

- Was the Km determined against ATP or the peptide? I suggest to explicitly state that.

Response: The Kfor the substrate peptide AQT0490 has been determined at a fixed saturating ATP concentration (5.0 mM MgATP).

- The specific activity of D321E is 20-fold higher and that of E322K is 1000 fold lower than that of wild type. These results seem different from those of others who did not observe reduced activity of Erk2 E to K mutant (for example Taylor et al., PNAS, 2019). These observations should be better described and seriously discussed.

Response: We have not compared in detail our results on D321N and E322K activity with those reported by Taylor et al. (2019) because it is difficult to compare kinetic parameters calculated by a continuous assay with the results obtained from an end-point activity assay, in different assay conditions. However, in Taylor et al. (PNAS 2019, Fig. 1F and G) D321E has not been reported, a similarity between D321N and wild-type activity and a decrease in E322K activity are reported, like we observed for these two variants. A comment about the D321G and E322K variants has been added at p. 13, line 462-464, at p. 16, line 542-553 and at p. 17, line 588-591.

Reviewer 2 Report

The authors of this manuscript investigated the impact of a series of variants on ERK2 trough a series of experimental techniques. The study is interesting and important as having the biophysical characterization of somatic variants in cancers is the first steps towards the understanding of oncogenic mechanisms. Prior to publication, I have a few questions that I believe should be addressed by the authors:

1) Could the authors provide an estimation of the errors associated with the temperatures for all variants mentioned in Table 1/3 (melting temperature and temperature of maximal activity)?

2) The statement, "These results indicate that the negatively charged residues in the CD-site are involved in the stabilization of the native state," appears to be inconsistent with the data. For instance, the mutation D321V, which replaces the negative residue in the CD-site with a hydrophobic residue, actually leads to an increase in stability, suggesting the marginal role of this residue in stability. Can the authors clarify this contradiction?

3) I think that the functional effects of variants are not well explained. Authors reported different experiments but the discussion on why these variants are related to oncogenic properties is poorly explained. I would suggest to separate discussions from results and add explanation by discussing each variants or class of variants. For example E322K increase the activity of ERK2, at the same time is the variant that impacts more the stability leading to an important drop in Tm. Data suggests that E322K lead to the resistance to Ravoxertinib but experiments from authors do not find a significants change in ERK2/Ravoxertinib affinity. It could be good to summarize for example for this variant all the data and provide a global interpretation of the variant role.  

4) Additionally, in the discussion section, it would be valuable for the authors to relate and compare their findings with previous investigations, such as the work by Taylor et al. (PNAS) that extensively characterizes E322K and D321N variants, as well as the study by Brenan et al. (Cell Reports, 2016) that should also be cited.

Minors

In Supplementary Material some + should be ±

Sometimes the ERK inhibitor is GD-C0994 instead to be labeled with GDC-0994

---

Author Response

The authors of this manuscript investigated the impact of a series of variants on ERK2 trough a series of experimental techniques. The study is interesting and important as having the biophysical characterization of somatic variants in cancers is the first steps towards the understanding of oncogenic mechanisms. Prior to publication, I have a few questions that I believe should be addressed by the authors:

We thank the Reviewer for the careful reading of the manuscript and for all the comments that helped in improving the quality of the manuscript.

Point 1: Could the authors provide an estimation of the errors associated with the temperatures for all variants mentioned in Table 1/3 (melting temperature and temperature of maximal activity)?

Response 1: according to the instrument supplier Jasco, the temperature control accuracy of the Peltier system we used is +/- 0.1°C; the temperature accuracy monitored during the experiments, i.e., the difference between the displayed temperature and the actual cell temperature, is +/- 0.5°C.

Point 2: The statement, "These results indicate that the negatively charged residues in the CD-site are involved in the stabilization of the native state," appears to be inconsistent with the data. For instance, the mutation D321V, which replaces the negative residue in the CD-site with a hydrophobic residue, leads to an increase in stability, suggesting the marginal role of this residue in stability. Can the authors clarify this contradiction?

Response 2: we agree with the referee’s observation. Mutation of the negatively charged residues D321 or E322 influences the native state stability, either increasing or decreasing. The sentence has been modified accordingly at p.16, line 541-553.

Point 3: I think that the functional effects of variants are not well explained. Authors reported different experiments but the discussion on why these variants are related to oncogenic properties is poorly explained. I would suggest to separate discussions from results and add explanation by discussing each variants or class of variants. For example E322K increase the activity of ERK2, at the same time is the variant that impacts more the stability leading to an important drop in Tm. Data suggests that E322K lead to the resistance to Ravoxertinib but experiments from authors do not find a significants change in ERK2/Ravoxertinib affinity. It could be good to summarize for example for this variant all the data and provide a global interpretation of the variant role.

Response 3: as far as concerns the relation between mutation and oncogenic properties, this point has been partly addressed in the Conclusions. Our study has been carried out on the pure protein variants to analyse the impact of the point mutation on the structural and biochemical properties of ERK2 in vitro, in experimental conditions very different from those utilized in the cells. Thus, the results concerning drug resistance in the cell for each variant may not be directly related to inhibitor binding affinities measured in vitro. However, as reported in Smorodinsky-Atias, et al., 2020 (doi.org/10.3390/cells9010129), “For most of the mutations that cause resistant to inhibitors the mechanism of action is not known and for many of them even the effects on Erks’ conformation and catalytic properties are yet to be revealed”. We have not separated Discussion and Results section to avoid changing the general organization of the manuscript that has been reviewed also by other referees.

Point 4: Additionally, in the discussion section, it would be valuable for the authors to relate and compare their findings with previous investigations, such as the work by Taylor et al. (PNAS) that extensively characterizes E322K and D321N variants, as well as the study by Brenan et al. (Cell Reports, 2016) that should also be cited.

Response 4: Our results are indeed complementary to what reported by Taylor et al. (2019) and by Brenan et al., (2016). However, we have not compared in detail our results on CD-site variants activity with those reported by other authors because it is difficult to compare kinetic parameters calculated with a continuous assay with the results obtained by an end-point activity assay, in different assay conditions, or structural properties in solution carried out on pure recombinant proteins. The study of Brenan et al. (2016), has been reported at p. 5, line 147-149 of the revised version and the relative reference has been added (ref [29]).

Minors

Point 5:  In Supplementary Material some + should be ±

Response 5: it has been corrected.

Point 6: Sometimes the ERK inhibitor is GD-C0994 instead to be labeled with GDC-0994

Response 6: it has been corrected.

Reviewer 3 Report

In this study the authors provide a thorough biochemical and biophysical analysis of point mutations in the ERK2 CD site that are found in human cancers. Previous studies have found that the CD site not only interacts with signaling partners but affects global changes in ERK2 structure and function, and this study provides an unparalleled analysis of the effects of mutations within the CD site.

Overall, the study is well-designed with appropriate controls and statistics. The results will help further understanding of CD site and overall ERK2 structure and function.

This study is suitable for publication. Only one major point and the several minor points listed below should be addressed.

The only major point to be addressed is the relative phosphorylation of the proteins. Co-expression of either a phosphatase or active MEK might be assumed to give similar levels of P-ERK but for such a thoroughly quantitative comparison of various mutants this should be confirmed experimentally (i.e. Western blot or mass spec).

Minor points…

-        Figure 1: indicate structure is of inhibitor bound ERK2

-        Figure 2:

o   color sidechain sticks with different colors for elements O, N, etc for clarity

o   Label F site

o   Activation loop color labeling appears to be incorrect. T and Y are within the activation loop but that region is not labeled green

-        Line 131: D321N and E322K NP-ERK2 stability were studied in ref 25 with similar results as this study. While this study provides a considerably more in-depth analysis it would be worthwhile to note that a previous study found something similar for two of the point mutants (i.e. a -5 and +5 degree change in thermal stability for the two mutants).

-        Figure 4: Label NP and P ERK2 for clarity

-        Figure 5: Label NP and P ERK2 for clarity

-        Comment more extensively on the D321G kinase activity since it is quite fascinating. Could this be due to the C-terminal region no longer folding correctly in the presence of a flexible residue? Almost no specific activity is quite dramatic.

-         Table 3: Were these measured in increments of 5 degrees? If so, this should be indicated.

-        Line 623: The D321N affinity is higher than WT but the sentence states that there is considerable disruption.

-        Line 631: M should be uM.

-        Figure 9: Specify your structure is bound to inhibitor in legend and provide a color key for easier interpretation by readers.

-        Line 694: Should it just be inversion or instead inversion/neutralization?

Overall English is readable but certain areas do not use proper grammar.

Author Response

In this study the authors provide a thorough biochemical and biophysical analysis of point mutations in the ERK2 CD site that are found in human cancers. Previous studies have found that the CD site not only interacts with signaling partners but affects global changes in ERK2 structure and function, and this study provides an unparalleled analysis of the effects of mutations within the CD site.

Overall, the study is well-designed with appropriate controls and statistics. The results will help further understanding of CD site and overall ERK2 structure and function.

This study is suitable for publication. Only one major point and the several minor points listed below should be addressed.

We thank the Reviewer for the careful reading of the manuscript and for all the comments that helped in improving the quality of the manuscript.

Point 1: The only major point to be addressed is the relative phosphorylation of the proteins. Co-expression of either a phosphatase or active MEK might be assumed to give similar levels of P-ERK but for such a thoroughly quantitative comparison of various mutants this should be confirmed experimentally (i.e., Western blot or mass spec).

 Response 1: A figure reporting the comparison between NP- and P-ERK2 variants by Western blot has been added as Supplementary Figure S1. We have not performed a quantitative comparison since the analysis was carried out on recombinant pure proteins and  a difference in the intensity of the bands may be due to a difference in the reactivity of the antibodies, raised against the wild-type, with the variants.

Minor points

Point 2: Figure 1: indicate structure is of inhibitor bound ERK2

Response 2: in the Figure 1 legend it has been explained that the pdb 4zzn corresponds to the human ERK2      bound to an inhibitor, not shown in the figure, whose structure has reported in the reference [12] (Ward et al., 2015).  

Point 3: Figure 2: color sidechain sticks with different colors for elements O, N, etc for clarity.

Response 3: Figure 2 has been modified, according to Reviewer suggestion.

Point 4: Label F site

Response 4: F-site has been labelled in Figure 2, according to Reviewer suggestion.

 Point 5: Activation loop color labeling appears to be incorrect. T and Y are within the activation loop but that region is not labeled green

Response 5: Figure 2 has been modified according to Reviewer suggestion.

Point 6: Line 131: D321N and E322K NP-ERK2 stability were studied in ref 25 with similar results as this study. While this study provides a considerably more in-depth analysis it would be worthwhile to note that a previous study found something similar for two of the point mutants (i.e. a -5 and +5 degree change in thermal stability for the two mutants).

Response 6: this point has been addressed at p. 5, line 150-152 and at p. 16, line 542-553, according to Reviewer suggestion.

Point 7: Figure 4: Label NP and P ERK2 for clarity

Response 7: Figure 4 (Figure 3 in the revised version) has been modified, according to Reviewer suggestion.

Point 8: Figure 5: Label NP and P ERK2 for clarity

Response 8: Figure 5 (Figure 4 in the revised version) has been modified, according to Reviewer suggestion.

Point 9: Comment more extensively on the D321G kinase activity since it is quite fascinating. Could this be due to the C-terminal region no longer folding correctly in the presence of a flexible residue? Almost no specific activity is quite dramatic.

Response 9: a comment about the impact of substitution of aspartate with glycine in D321G on the catalytic activity and on the biophysical properties of this variant has been added at p. 13 , line 462-464, at p. 14, line  494-495 and at p. 16, line 548-550.

Point 10: Table 3: Were these measured in increments of 5 degrees? If so, this should be indicated.

Response 10: the temperature interval for temperature dependence activity has been reported in detail in Materials and Methods, section 2.7.

Point 11: Line 623: The D321N affinity is higher than WT but the sentence states that there is considerable disruption.

Response 11: with the sentence “These results confirmed structural changes in D321N within the CD-site that interfere with this docking interaction” we do not mean disruption but structural changes that may affect binding interactions. As a matter of fact, D321N binds with a higher affinity the pepMEK2 peptide and does not bind at all the KIM peptide of DUSP6. The sentence has been changed to avoid confusion, in agreement with the referee consideration.

Point 12: Line 631: M should be uM.

Response 12: it has been corrected.

Point 13: Figure 9: Specify your structure is bound to inhibitor in legend and provide a color key for easier interpretation by readers.

Response 13: Figure 9 (Figure 8 in the revised version) has been modified, according to Reviewer suggestion.

Point 14: Line 694: Should it just be inversion or instead inversion/neutralization?

Response 14: the sentence has been modified, according to the Referee observation.

Round 2

Reviewer 1 Report

No comments, beyond those of the first round.